# Analysis of Intensities Using Inertial Motion Devices in Female Soccer: Do You Train like You Compete?

**DOI:** 10.3390/s22082870

**Published:** 2022-04-08

**Authors:** Juan M. García-Ceberino, Ana Bravo, Ernesto de la Cruz-Sánchez, Sebastián Feu

**Affiliations:** 1Faculty of Humanities and Social Sciences, University of Isabel I, 09003 Burgos, Spain; juanmanuel.garcia.ceberino@ui1.es; 2Optimization of Training and Sports Performance Research Group (GOERD), University of Extremadura, 10003 Cáceres, Spain; 3Sport Extremadura, Sports Club, 06006 Badajoz, Spain; abravog62@gmail.com; 4Public Health and Epidemiology Research Group, University of Murcia, 30003 Murcia, Spain; erneslacruz@um.es; 5Faculty of Education, University of Extremadura, 06006 Badajoz, Spain

**Keywords:** external intensity, female player, internal intensity, official match, training session

## Abstract

Intensity research in female soccer is limited. This study aimed to investigate whether female professional soccer players train with external and internal intensities similar to those recorded in real competition. The specific players’ position, the game situation and training task type were analyzed in a total of 18 female players (26.25 ± 3.89 years). The empirical, descriptive and associative study was structured into two parts. Part 1: characterizing the training sessions (*n* = 13) and official matches (*n* = 3) using the Integral Analysis System of Training Tasks. The association between sports planning variables was evaluated using adjusted standardized residuals from contingency tables, Chi-Square and Fisher tests, as well as the Phi and Cramer’s V coefficients. The main findings show that the coach and/or physical trainer predominantly planned training sessions using small-sided games, which integrate physical fitness and tactical-technical behaviors of the game and imply a medium-high subjective external intensity (20.63 ± 5.79 points). The subjective external intensity of the matches was very high (30.00 ± 0.00 points). Part 2: quantifying the external and internal intensity through the inertial motion devices and heart rate monitors. Differences in the intensities according to the type of session (training session and match), specific position of the players, game situation and type of the training task were assessed through different statistical tests. By specific position (Kruskal–Wallis H and one-factor ANOVA tests), defenders performed fewer accelerations/min and decelerations/min, while they recorded higher heart rates in training sessions and official matches. In contrast, the wingbacks performed higher accelerations/min and decelerations/min in training sessions and official matches. The wingers had the lowest heart rate in official matches. Regarding the game situation (Kruskal–Wallis H test) measured during training sessions, the unopposed tasks recorded higher accelerations/min and decelerations/min, while the small-sided games and full games recorded higher values in the rest of the intensities (both subjective and objective). With regard to the type of training task (Kruskal–Wallis H test), the simple application exercises recorded higher accelerations/min and decelerations/min. Distance in meters/min was greater in the complex application exercises. High-intensity activity/min and player load/min were higher in the simple specific game. In addition, modified sport and real game recorded higher subjective external intensity*min, sprints/min and heart rate. Furthermore, training sessions differed statistically (Mann–Whitney U test) from official matches in terms of subjective intensity and the objective external and internal intensity variables weighted by minutes. For all these reasons, female players do not train (training sessions) as they compete (official matches). The use of inertial motion devices has made it possible to quantify intensities during training sessions and real competition in soccer.

## 1. Introduction

The practice of female soccer has increased during recent years. One aspect of the increase in this sport is the fact that around 30 million female senior and youth players worldwide are involved in soccer club [1]. In Spain, female soccer has become one of the most practiced sports, extending to all levels (educational, recreational and competitive), and it will continue to increase [2].

At the competitive level, the rise in female soccer has led to further research on those small-sided game (SSG) formats that coaches and trainers employ in order to train the tactical-technical behaviors and physical fitness of female soccer players [3]. With respect to the soccer sport, Hill-Haas et al. [4] define SSGs as game-based training, in which game spaces are modified, with adapted rules and a smaller number of players than traditional soccer matches. The SSG use allows players to face situations with technical, tactical, physical and physiological demands similar to those encountered in real competition [5]. Therefore, they are commonly used in the context of sports training and youth player development programs [6].

The physical-physiological demands of female soccer players can be measured using intensity monitoring (this replaces the term “workload” [7,8,9]). The intensity monitoring represents a quantitative value of the work performed by players during a training session or competitive match [10]. In this regard, the intensities are classified as follows: the external intensity (eTL) and internal intensity (iTL). The eTL refers to the physical demands to which the players are subjected, while iTL refers to the physiological demands that a stimulus planned (physical demands) generates in the players [11]. Objectively, the eTL is divided into kinematic and neuromuscular variables. The kinematic variables analyze the movements/displacements and their intensities: distance in meters, accelerations, decelerations, speeds, high intensity activities, etc. [12]. The neuromuscular variables analyze the forces exerted by the players as a consequence of the interaction with gravity and opponents: player load (PL), impacts, etc. The PL is the most predictive variable of objective eTL [13]. On the other hand, the heart rate (HR) measures the intensity, objective iTL, of sports practice [14]. The above-mentioned intensities can be quantified through the inertial motion devices and HR monitors synchronized with them [15].

Knowledge of the physical-physiological demands of sport competition is one of the factors that will allow coaches and/or physical trainers to plan training sessions with intensities similar to those of competition [16]. With regard to the training sessions, the intensities can be modified by varying the formal and structural elements (constraints) of the training tasks, such as the number of the players involved, the size of the game spaces, the presence or not of the goalkeeper, the rules modification, etc. [4], in order to obtain an adequate sports performance. This type of format refers to SSGs, while the more realistic game situations (namely, soccer, 11 vs. 11) are known as full games [17]. Considering the physical-physiological impact of the SSG constraints, in general, a larger game space implies a greater distance covered at higher speed and more sprints, while training tasks with fewer players are more fatiguing because the participation is greater [5]. Therefore, training sessions must be structured, systematic and organized in advance by the coaches and/or physical trainers, in order to achieve successive learning and optimal physical preparation for the competition [18]. There is a cause–effect relationship between training sessions and competition.

Training sessions are the responsibility of coaches and/or physical trainers, as they clearly act on the design of the training tasks, affecting the eTL and iTL of the players [18]. In this regard, the sports planning process can be optimized through the Integral Analysis System of Training Tasks, SIATE, which has five fundamental characteristics: universality, normalization, modularity, flexibility and adaptability to all invasive sports in which it is applied. It is a methodological system that allows for the recording and subsequent analysis of the different (pedagogical, organizational and subjective external intensity) factors that affect the sports training process [19]. These factors have been analyzed in several studies on soccer [20,21,22]. The SIATE is also a useful technique for subjectively quantifying eTL, and it is correlated with other objective and subjective intensity techniques in formative-level soccer [11,23]. The coaches and/or physical trainers also have another technique for recording subjective iTL, called the Borg scale [24].

The monitoring of players’ eTL and iTL has become an interesting topic for researchers during training sessions and competition [25,26], and allows one to analyze the differences and similarities. Inertial motion devices allow for the study of the influence of SSGs on physical fitness, technique and changes of direction [3], the comparisons of physical demands between SSGs and competition [5] and the study of the effects of a program of SSGs on the subjective perception of effort [27]. To our knowledge, few studies have compared the physical-physiological demands of female soccer players between training and competition. Thus, we aimed to study whether female professional soccer players train with intensities (eTL and iTL) similar to those recorded in real competition. The specific position of the players, the game situation and type of training task were also analyzed.

## 2. Materials and Methods

### 2.1. Study Design

This study was framed within an empirical, descriptive and associative strategy of a comparative and cross-sectional type [28]. It also comprises a meso-cycle of the competition period of the 2020/2021 season of the Spanish National Female’s Soccer League. The meso-cycle was composed: first week (3 training sessions + official match), second week (1 training session + official match), third week (3 training sessions), fourth week (3 training sessions) and fifth week (3 training sessions + official match). The average number of players analyzed per training session was 13. In the official matches, the 10 field players who started the matches were analyzed. Figure 1 shows the days of data collection.

### 2.2. Population and Sample

A total of 18 senior female players (26.25 ± 3.89 years) from a professional soccer team registered in the Iberdrola First Division (Spanish National Female’s Soccer League) in the 2020/2021 season participated in the study. All active, non-injured players on the senior team participated in the study. Likewise, Table 1 describes the characteristics of the players according to their specific position.

In order to guarantee the ethical considerations for scientific investigations with human beings, the study was conducted according to the ethical guidelines of the Declaration of Helsinki of 1975 (it modified in subsequent years), and Organic Law 3/2018 of 5 December on the protection of personal data and guarantee of digital rights (BOE, 294, 6 December 2018).

### 2.3. Variables and Instruments

The following variables and instruments were selected for this study (Table 2):

### 2.4. Procedure

#### 2.4.1. Authorizations

The approval of the University’s Extremadura Bioethics Committee (Code: 09/2018) was obtained. Then, an informative meeting was held with the president, coach and physical trainer of the Club, who accepted the study. Furthermore, the female players signed a written informed consent.

#### 2.4.2. Protocols

The researchers explained to the female players how to turn on, operate and turn off the inertial motion devices at the beginning of the study. In addition, the physical trainer provided the anthropometric measurements.

The following protocol was followed for each training session and official match:Prior to each training session, the coach or physical trainer provided the researchers with the session for the categorization of the training tasks using SIATE observation sheet [19].Assistance of the researchers at the stadium 30–40 min and 1 h before the start of training sessions and official matches, respectively.Once the inertial devices were turned on and calibrated, each female player put on her inertial device (numbered), together with the HR monitor. Then, the recording began. The inertial devices were placed in anatomical harnesses during training sessions and official matches.At the end of training sessions and official matches, the female players were responsible for turning off the inertial devices. In this way, the researchers only had to intervene to collect them in their briefcase.

#### 2.4.3. Data Extraction (WIMU Pro^TM^ Inertial Device)

After each training session and official match, the data were downloaded through the SPRO^TM^ v964 software (RealTrack System, Almería, Spain), which allows the quantitative data to be exported to the statistical program. This inertial motion device was integrated through different sensors to record the data: a global position system or GPS, four accelerometers, a magnetometer, a gyroscope, an ultra-wideband (UWB) chip among others. The frequency for the chip’s signal was 18 Hz. It has strong validity and reliability for sports analysis [15,31,32].

### 2.5. Statistical Analysis

The criterion assumption tests indicated the use of non-parametric tests (*p* < 0.05) for hypothesis testing, with the exception of the eTL variables acc/min, dec/min, HIA/min and PL/min (*p* > 0.05) when analyzing differences according to the specific position of the players in official matches [33].

First, a descriptive analysis was performed to determine the frequency and percentage of the different categories of each group of variables (pedagogical, organizational and subjective eTL) recorded using SIATE observation sheet [19]; this defined the training sessions and official matches. The mean and standard deviation were also calculated for the objective intensity variables. Then, the Adjusted Standardized Residuals (*ASR*) of the contingency tables were analyzed to find the differences (*ASR* > |1.96|) between the variables studied. In addition, the strength of association was calculated. For this, the Chi-Square test (*X*^2^), Fisher’s exact test (*f*), and Cramer’s Phi (*Φc*) and Cramer’s V (*Vc*) coefficients were used [33], attending to the values proposed by Crewson [34].

On the other hand, the inferential analyses of the intensities according to the type of session (training and official match), specific position, game situation and type of training task were performed using Mann–Whitney U (2 categories), Kruskal–Wallis H (>2 categories) and one-factor ANOVA (>2 categories) statistical tests. In the official matches, it was not possible to perform the inferential analysis according to game situation and training task type because there was only one game situation (11 vs. 11, full game category) and one type of training task (competition category) (Table 3). It was necessary to select the intensities per minute in order to establish the differences, since the training tasks present different duration times.

Finally, the effect size was calculated using Rosenthal’s r test (for the Mann–Whitney U test), the Epsilon-Squared (*E*^2^*_R_*) coefficient (for the Kruskal-Wallis H test) and the Partial Eta-Squared (*ηp*^2^) index (for the one-factor ANOVA test) [33,35].

SPSS v25.0 was used for statistical analysis (IBM Corp. Released 2017. IBM SPSS Statistics for Windows, Version 25. IBM Corp. U.S.A., Armonk, NY, USA). The level of significance was *p* < 0.05. The graphics were made with GraphPad Prism v8.0.1 (Graphpad Inc., La Jolla, CA, USA, EE. UU.).

## 3. Results

### 3.1. Characterization of Training Sessions and Official Matches (Study Part 1)

Table 3 and Table 4 show the percentages, *ASR* and the association between the categories of the pedagogical, organizational and eTL variables (SIATE observation sheet) according to the type of session: training session and official match.

### 3.2. Differences of the Intensities according to Specific Position, Game Situation and Task Type (Study Part 2)

Figure 2 details the descriptive and inferential results of the training sessions according to three groups of variables: specific player position, game situation and training task type (training means).

Descriptive and inferential results of the official matches (competition) only according to specific player position are shown in Figure 3. Descriptive results of the full game and competition categories detail: subjective eTL*min = 1160.44 ± 298.79; m/min = 91.28 ± 11.64; acc/min = 31.15 ± 2.60; dec/min = 31.14 ± 2.61; HIA/min = 2.20 ± 1.84; sprint/min = 7.11 ± 14.87; PL/min = 1.28 ± 0.24; HR_avg_ = 152.51 ± 18.30; and relative HR% = 80.58 ± 10.03.

Table 5 shows the pairwise comparisons (*p* < 0.05) according to the intensity variables analyzed. The first category of the pairwise comparisons has the highest average rank.

### 3.3. Differences of the Intensities according to the Type of Session (Study Part 2)

Table 6 shows the inferential results for the eTL, weighted per minute, and iTL variables according to the type of session: training session and official match.

## 4. Discussion

The study aimed to investigate whether female professional soccer players train with intensities (eTL and iTL) similar to those registered in real competition, i.e., in official matches. Regarding the training sessions, the main findings indicate a predominance of tasks played with opposition (SSGs) to train tactical-technical and physical fitness contents, and that caused a medium-high subjective eTL. Intensities also differed statistically in training sessions and official matches according to the position-specific player, game situation and type of task. Furthermore, subjective eTL*min (SIATE sheet), objective eTL weighted per minute and iTL were not similar in training sessions and matches. The SSGs should be used predominantly in training sessions because they cause the closest intensities to competition. In addition, it is necessary to adjust the task type (constraints) to the specific position.

### 4.1. Characterization of Training Sessions and Official Matches (Study Part 1)

The organization and systematization of soccer training requires strategies, often based on the principles of the game [36]. With regard to to the training sessions designed by the coach and/or physical trainer, the analysis of the pedagogical, organizational and subjective eTL variables [19] indicated that 32.00% of the tasks were warm-up and 18.70% were physical fitness. Regarding soccer-specific content, mixed tasks stood out (25.80%), followed by attack (11.80%) and defense (11.60%) tasks. Likewise, the adulatory model indicated by Ibáñez [37] was fulfilled, where the game phases are combined and the contents of attack and defense are balanced. Similarly, there was a predominance of attacking (14.50%) and defensive (9.20%) group tactical behaviors and of attacking (10.00%) team tactical behaviors over technical execution skills, with an organization of simultaneous participation of the players (76.00%).

For the training of these contents, 58.20% were tasks with opposition, while 38.50% were tasks without opposition. In addition, 41.80% were application exercises (simple and complex), while the rest of the activities were situations played, with 25.80% being specific complex games and 17.40% full games. Therefore, a higher percentage of activities played (compared to exercises) was observed, their use being more appropriate since they are the closest to real competition [38]. Along this line, SSGs are used in female soccer because players train with demands (tactical-technical behaviors, physical fitness, decision making, etc.) close to competition, both in formative-level soccer [3,5] and elite-level soccer [39]. On the other hand, the official matches were categorized as full games of equal numerical oriented to the competition (100.00%).

In addition, movements/displacements with repetition in spaces predominated in the training sessions (68.30%). This fact, together with a high number of simultaneous players and high task density, caused the medium-high subjective eTL (*M* = 20.63) [19,40]. On the other hand, the categorization of the official matches indicated a very high subjective eTL (*M* = 30.00). In this regard, a previous research on formative-level soccer showed that subjective eTL correlates positively with the objective eTL (PL) and iTL (HR) [11,23].

### 4.2. Differences of the Intensities according to Specific Position, Game Situation and Task Type (Study Part 2)

Regarding the specific position of the player, there were no significant differences in subjective eTL*min in both training sessions and official matches. In contrast, there were significant differences in acc/min, dec/min, PL/min and HR in the training sessions. Wingbacks registered higher acc/min and dec/min, while wingers and strikers recorded higher PL/min, and defenders higher HR values. In the official matches, there were also significant differences in acc/min, dec/min and HR_avg_ prevailing at the same specific positions in each intensity. In a previous study on sub-elite female soccer players [26], the authors analyzed positional differences during competitive matches. Midfielders recorded greater total distance in m/min (*p* < 0.05) and PL/min, while strikers recorded greater distance in HIA/min (*p* > 0.05). Likewise, defenders moved less distance, but they spent a greater percentage of time in the high-intensity zone of HR per minute. Thus, the results were similar to those obtained in this study. Knowing the eTL and iTL of official matches provides a description of the physical-physiological demands to which female players are subjected. These findings will help coaches and/or physical trainers to avoid injuries [41,42] and to plan training sessions that will improve the performance of position-specific players and perform at a better level in matches.

Moreover, significant differences were observed in all intensities taking into account the game situation. The without opposition tasks, categorized as simple application exercises, recorded higher acc/min and dec/min, while SSGs and full games recorded higher levels in the rest of the intensities studied. In this regard, Nevado-Garrosa and Suárez-Arrones [5] stated in their study on U13 female soccer players that the increase in the number of players and game spaces implied more demanding physical demands, with players moving greater total distances/m and at greater speeds. Thus, the number of sprint/m was also higher, which could be due to the increased game space in the SSGs and sport [5,39]. Mara et al. [43] obtained that elite female soccer players performed grater relative sprint distances during full games than in the SSGs; however, SSGs caused a greater number of total acceleration efforts. The greater number of accelerations and decelerations caused by without opposition tasks could be due to the movement/displacement involved in their execution, as well as to the durations of these movements [44]. Therefore, during these game situations (e.g., 1 vs. 0, 2 vs. 0, 3 vs. 0…), soccer players move repeatedly for a few meters, accelerating to the maximum at the beginning of the tasks without being a sprint (<21 km/h) and then decelerating rapidly [45]. Likewise, numerical equality SSGs stood out above the rest in the PL/min, which could be due to changes in direction, accelerations and decelerations.

In general, increasing the game space per player means more physical demands [5]. Coinciding with this study, previous research on sub-elite female soccer players [46,47] stated that they seem to play more intensively (HR) in larger spaces, i.e., full games. In contrast to this, other authors [4,43] stated that the tasks with fewer players (SSGs) tend to be more fatiguing (>HR) than full games. Furthermore, Jastrzebski et al. [48] obtained that there were no physiological-HR differences between male and female professionals when playing 4 vs. 4 SSG soccer. These authors also stated that female players have a lower aerobic potential that determines a lower total distance cover during SSGs, while male players cover greater distances running at high speed.

Considering the task type (training means), there were also significant differences in all intensities. Complex application exercises recorded higher distance in m/min. In addition, the simple specific games, categorized as SSGs, stood out notably over the rest in the PL/min and HIA/min. The eTL*min, sprint/min and HR variables were higher in adapted games (SSGs) and sport. These findings indicate that SSGs are the closest to the physical-physiological demands of the sport of soccer (official matches) [3,39] because they are contextualized to the reality of the game and competition.

### 4.3. Differences of the Intensities according to the Type of Session (Study Part 2)

Significant differences showed that female players do not train with the same intensities as those recorded in official matches. In this regard, the task type (training means) designed by the coach and/or physical trainer caused a higher number of acc/min and dec/min in the training sessions, which in turn determined a higher PL/min. However, the intensity was higher in the official matches, as evidenced by the higher values recoded in the rest of the eTL and iTL (HR) variables. In this regard, Ohlsson et al. [42] also compared training sessions with competition in elite female soccer players. These authors demonstrated higher HR values during competition compared to training sessions, indicating a greater demand on the iTL of the female players during the matches. Therefore, the type of training task [11,45] and game spaces [5,46,47] modified the neuromuscular, kinematic and HR variables. In our study, despite the majority use of SSGs, 41.80% were application exercises, which could indicate the reasons why the female players did not train with the same intensities as those recorded in official matches (competition). Thus, the results suggest the need for including more high-intensity tasks (SSGs) and fewer application exercises during training sessions.

### 4.4. Limitations and Practical Applications

One of the limitations of this study that must be taken into account is that not all the female players participated in all training sessions and official matches. Furthermore, depending on the SSGs used by the coach and/or physical trainer, female soccer players will be able to reproduce more or less accurately the demands of competition. In this regard, Nevado-Garrosa and Suárez-Arrones [5] observed that 7 vs. 7 SSGs reproduce similar demands to competition.

It has been shown that the modification of pedagogical, organizational and subjective external intensity variables [19,45] that define training tasks causes modifications in players’ eTL and iTL demands. Thus, future research should focus on the study of these variables, expanding the possibility of comparing and justifying the results. Likewise, they should analyze the physical-physiological demands caused by female soccer players according to the specific position in order to plan training sessions to improve their performance in official matches.

This study delved into the match demands of professional female soccer players and can help coaches and/or physical fitness professionals design specific training sessions to improve their performance in competition. Likewise, the use of motion sensors will help sports professionals in monitoring intensities.

## 5. Conclusions

The coach and/or physical trainer mostly used the SSG format to train the tactical-technical behaviors and physical fitness of female players. SSGs are the ones that cause physical-physiological demands similar to competition; therefore, SSGs are recommended in elite female soccer. However, the results show that training sessions and official matches did not register similar intensities. This fact could be due to a high use of without-opposition exercises and the dimensions of the game spaces in the training sessions. The tasks played, such as SSGs, with repetition of large game spaces cause intensities closer to competition.

## Figures and Tables

**Figure 1 sensors-22-02870-f001:**
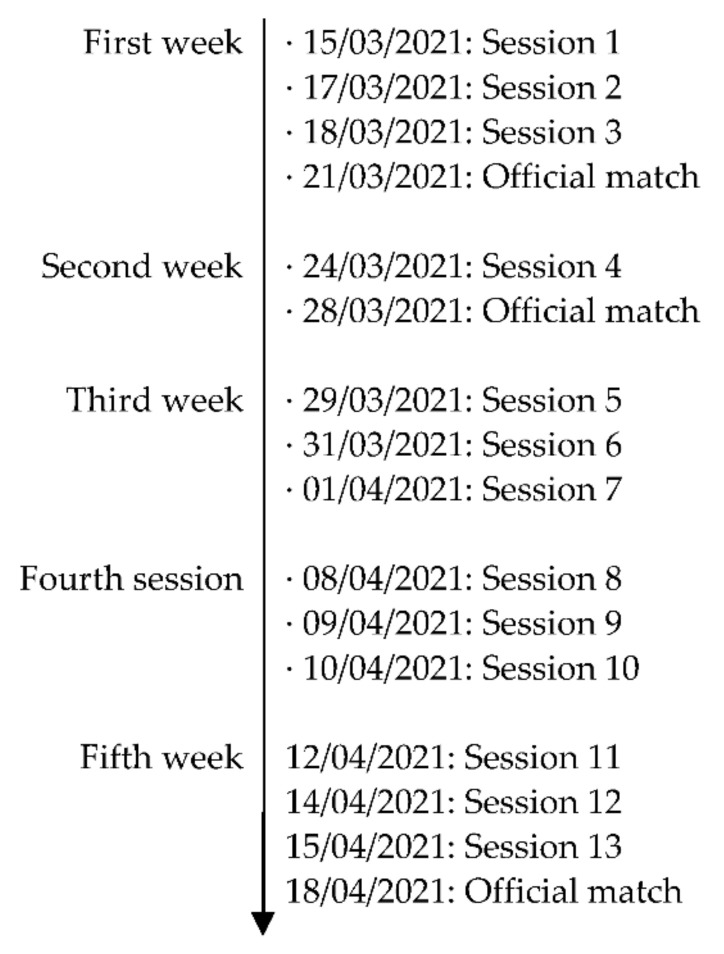
Chronology of the competitive meso-cycle.

**Figure 2 sensors-22-02870-f002:**
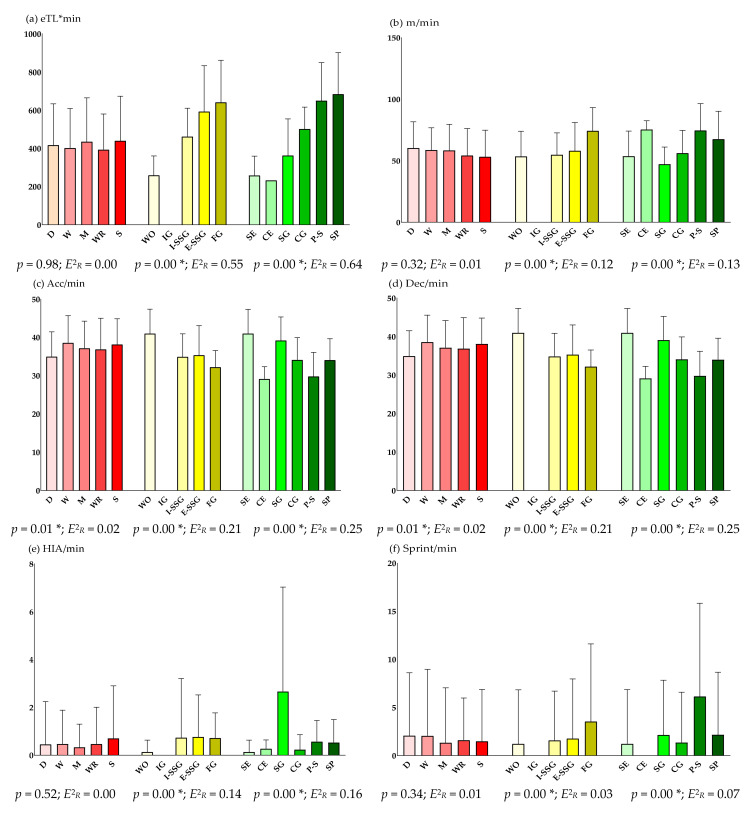
Descriptive and inferential results of the training sessions according to the specific player position (red color), game situation (yellow color) and training task type (green color). Note: *E*^2^*_R_* = Epsilon-Squared Coefficient; eTL = external Training Intensity; min = Minute; m = Meter; Acc = Acceleration; Dec = Deceleration; HIA = High-Intensity Activity; PL = Player Load; iTL = internal Training Intensity; HR = Heart Rate; avg = Average; D = Defender; W = Wingback; M = Midfielder; WR = Winger; S = Striker; WO = Without Opposition Task; IG = Individual Game; I-SSG = Inequality Small-Sided Game; E-SSG = Equality Small-Sided Game; FG = Full Game; SE = Simple Exercise; CE = Complex Exercise; SG = Simple Game; CG = Complex Game; P-S = Adapted Game; SP = Sport. * *p* < 0.05 (Kruskal–Wallis H).

**Figure 3 sensors-22-02870-f003:**
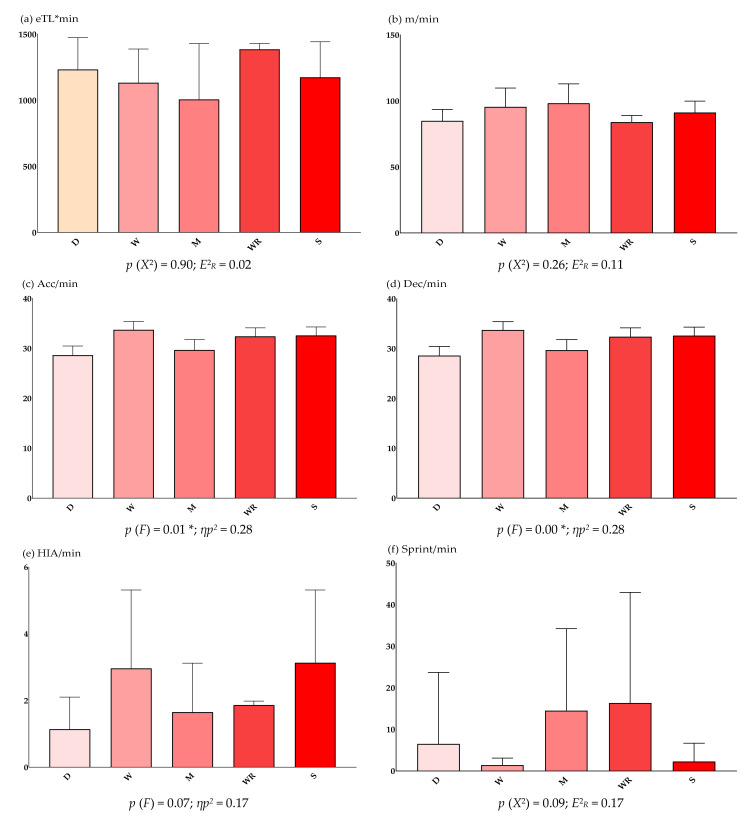
Descriptive and inferential results of the official matches according to specific player position. Note: *X*^2^ = Kruskal–Wallis H Test; *E*^2^*_R_* = Epsilon-Squared Coefficient; *F* = One-factor ANOVA Test; *ηp2* = Partial Eta-Squared Index; eTL = external Training Intensity; min = Minute; m = Meter; Acc = Acceleration; Dec = Deceleration; HIA = High-Intensity Activity; PL = Player Load; iTL = internal Training Intensity; HR = Heart Rate; avg = Average; D = Defender; W = Wingback; M = Midfielder; WR = Winger; S = Striker. * *p* < 0.05.

**Table 1 sensors-22-02870-t001:** Characteristics of the female players by specific position.

Age and Anthropometry	All	Defender	Wingback	Midfielder	Winger	Striker
	M ± SD					
Age (years)	26.25 ± 3.89	23.94 ± 1.00	25.29 ± 1.29	24.20 ± 4.16	24.14 ± 3.01	31.34 ± 1.57
Body mass (kg)	60.72 ± 7.30	70.53 ± 6.33	57.70 ± 3.45	61.04 ± 8.06	55.86 ± 3.01	58.62 ± 4.52
Stature (m)	1.66 ± 0.07	1.74 ± 0.04	1.57 ± 0.04	1.65 ± 0.03	1.62 ± 0.01	1.69 ± 0.05
BMI (kg/m^2^)	22.11 ± 1.98	23.21 ± 1.54	23.33 ± 0.39	22.40 ± 2.84	21.15 ± 1.01	20.49 ± 1.10

Note: M = Mean; SD = Standard Deviation; kg = Kilogram; m = Meter; BMI = Body Mass Index.

**Table 2 sensors-22-02870-t002:** Variables and data collection instruments.

Variable	Category	Instrument	Consideration
Pedagogical variables	Game Situation (GS)	SIATE observation sheet [19]	Content analysis of training tasks (categorical variables, less TTT variable) Three coders (researchers), experienced in the field of Sport Pedagogy and in the use of this instrument, performed the categorization of the tasks. Inter-coder reliability (Multirater _Kfree_) is proven [29,30]
	Presence of Goalkeeper (POG)		
	Game Phase (GP)		
	Type of Content (CONT-G)		
	Training Means (TM)		
	Level of Opposition (LO)		
Organizational variables	Total Task Time (TTT): min		
	Type of Participation (TP)		
Subjective eTL variables	Degree of Opposition (DO)		
	Density of the Task (DT)		
	% Simultaneous Performers (PSP)		
	Competitive Load (CL)		
	Game Space (GS)		
	Cognitive Implication (CI)		
	eTL*min	(eTL = DO + DT + PSP + CL + GS + CI)*min	
Objective eTL (kinematic variables)	Distance (m/min)	WIMU Pro^TM^ (Wireless Inertial Unit Movement) (RealTrack System, Almería, Spain)	Inertial motion device and HR monitor synchronized with it via Ant+ technology [15] (scale variables) These variables were only measured at the time of motor commitment (useful time)
	Accelerations/min (Acc/min)		
	Decelerations/min (Dec/min)		
	HIA/min		
	Sprints/min		
Objective eTL (neuromuscular variable)	PL/min		
Objective iTL variables	HR_avg_		
	Relative HR%		

Note: SIATE = Integral Analysis System of Training Tasks; eTL = external Training Intensity; min = Minute; m = Meter; HIA = High-Intensity Activity; PL = Player Load; iTL = internal Training Intensity; HR = Heart Rate; avg = Average.

**Table 3 sensors-22-02870-t003:** Percentages, *ASR* and association of the pedagogical and organizational variables.

Variable/Category (SIATE Observation Sheet)		Training Session			Official Match			Association		
		*n*	%	*ASR*	*n*	%	*ASR*	*X*^2^ (*p*)	*f* (*p*)	*Vc*
GS	With Opposition	265	40.00	5.6 ^†^	0	0.00	−5.6 ^†^	210.88 (0.00 *)	162.15 (0.00 *)	0.54
	Individual Game	-	-	-	-	-	-			
	Inequality SSG	201	30.40	4.6 ^†^	0	0.00	−4.6 ^†^			
	Equality SSG	103	15.60	3.0 ^†^	0	0.00	−3.0 ^†^			
	Full Game	93	14.00	−14.5 ^†^	49	100.00	14.5 ^†^			
POG ^1^	With Goalkeeper	253	38.20	−8.4 ^†^	49	100.00	8.4 ^†^	71.27 (0.00 *)	-	0.30
	Without Goalkeeper	409	61.80	8.4 ^†^	0	0.00	−8.4 ^†^			
GP	Attack	78	11.80	2.5 ^†^	0	0.00	−2.5 ^†^	117.45 (0.00 *)	109.08 (0.00 *)	0.41
	Defence	77	11.60	2.5 ^†^	0	0.00	−2.5 ^†^			
	Mixed	171	25.80	−10.8 ^†^	49	100.00	10.8 ^†^			
	Warm-Up	212	32.00	4.7 ^†^	0	0.00	−4.7 ^†^			
	Physical Fitness	124	18.70	3.3 ^†^	0	0.00	−3.3 ^†^			
CONT-G	AGTTB	96	14.50	2.9 ^†^	0	0.00	−2.9 ^†^	711.00 (0.00 *)	326.32 (0.00 *)	1.00
	DGTTB	61	9.20	2.2 ^†^	0	0.00	−2.2 ^†^			
	AGTTM	35	5.30	1.7	0	0.00	−1.7			
	ATTTB	66	10.00	2.3 ^†^	0	0.00	−2.3 ^†^			
	DTTTB	16	2.40	1.1	0	0.00	−1.1			
	ATTTM	12	1.80	1.0	0	0.00	−1.0			
	Warm-Up	212	32.00	4.7 ^†^	0	0.00	−4.7 ^†^			
	Physical Fitness	124	18.70	3.3 ^†^	0	0.00	−3.3 ^†^			
	Training Match	40	6.00	1.8	0	0.00	−1.8			
	Official Match	0	0.00	−26.7 ^†^	49	100.00	26.7 ^†^			
TM ^2^	Simple Exercise	264	40.00	5.6 ^†^	0	0.00	−5.6 ^†^	709.00 (0.00 *)	333.21 (0.00 *)	1.00
	Complex Exercise	12	1.80	1.0	0	0.00	−1.0			
	Simple Game	66	10.00	2.3 ^†^	0	0.00	−2.3 ^†^			
	Complex Game	170	25.80	4.1 ^†^	0	0.00	−4.1 ^†^			
	Adapted Sport	33	5.00	1.6	0	0.00	−1.6			
	Sport	115	17.40	3.2 ^†^	0	0.00	−3.2 ^†^			
	Competition	0	0.00	−26.6 ^†^	49	100.00	26.6 ^†^			
LO	Without Opposition	255	38.50	5.4 ^†^	0	0.00	−5.4 ^†^	33.59 (0.00 *)	44.30 (0.00 *)	0.22
	Static Obstacles	22	3.30	1.3	0	0.00	−1.3			
	With Opposition	385	58.20	−5.8 ^†^	49	100.00	5.8 ^†^			
TTT ^1^	≤10 min	41	6.20	1.8	0	0.00	−1.8	3.22 (0.07)	-	0.07
	>10 min	621	93.80	−1.8	49	100.00	1.8			
TP	Simultaneous	503	76.00	−3.9 ^†^	49	100.00	3.9 ^†^	15.16 (0.00 *)	19.23 (0.00 *)	0.15
	Alternative	104	15.70	3.0 ^†^	0	0.00	−3.0 ^†^			
	Consecutive	55	8.30	2.1 ^†^	0	0.00	−2.1 ^†^			

Note: *n* = Frequency; *ASR* = Adjusted Standardized Residuals; *X*^2^ = Chi-Square Test; *f* = Fisher’s Exact Test; *Vc* = Cramer’s V Coefficient; GS = Game Situation; SSG = Small-Sided Game; POG = Presence of Goalkeeper; GP = Game Phase; CONT-G = Type of Content; AGTTB = Attacking Group Tactical-Technical Behaviors; DGTTB = Defensive Group Tactical-Technical Behaviors; AGTTM = Attacking Group Technical-Tactical Moves; ATTTB = Attacking Team Tactical-Technical Behaviors; DTTTB; Defensive Team Tactical-Technical Behaviors; ATTTM = Attacking Team Technical-Tactical Moves; TM = Training Means; LO = Level of Opposition; TTT = Total Task Time; TP = Type of Participation. ^1^
*Φc* Coefficient (2 × 2 table) was used. ^2^ Simple and complex games are specific. ^†^ *ASR* > |1.96|; * *p* < 0.05.

**Table 4 sensors-22-02870-t004:** Percentages, *ASR* and association of the subjective eTL variables.

Variable/Category (SIATE Observation Sheet)		Training Session			Official Match			Association		
		*n*	%	*ASR*	*n*	%	*ASR*	*X*^2^ (*p*)	*f* (*p*)	*Vc*
DO	With Opposition	265	40.00	5.6 ^†^	0	0.00	−5.6 ^†^	100.10 (0.00 *)	97.93 (0.00 *)	0.37
	>3 or +Players	77	11.60	2.5 ^†^	0	0.00	−2.5 ^†^			
	>2 Players	45	6.80	1.9	0	0.00	−1.9			
	>1 Player	79	11.90	2.6 ^†^	0	0.00	−2.6 ^†^			
	Numerical Equality	196	29.60	−10.0 ^†^	49	100.00	10.0 ^†^			
DT	Walking	-	-	-	-	-	-	121.41 (0.00 *)	115.42 (0.00 *)	0.41
	Gentle Pace	169	25.50	4.1 ^†^	0	0.00	−4.1 ^†^			
	Intensity With Rest	298	45.00	6.2 ^†^	0	0.00	−6.2 ^†^			
	Intensity Without Rest	29	4.40	1.5	0	0.00	−1.5			
	High Intensity Without Rest	166	25.10	−11.0 ^†^	49	100.00	11.0 ^†^			
PSP	<20%	12	1.80	1.0	0	0.00	−1.0	15.16 (0.00 *)	16.31 (0.00 *)	0.15
	21–40%	43	6.50	1.8	0	0.00	−1.8			
	41–60%	90	13.60	2.8 ^†^	0	0.00	−2.8 ^†^			
	61–80%	14	2.10	1.0	0	0.00	−1.0			
	>81%	503	76.00	−3.9 ^†^	49	100.00	3.9 ^†^			
CL	Without Competition	240	36.30	5.2 ^†^	0	0.00	−5.2 ^†^	149.63 (0.00 *)	129.03 (0.00 *)	0.46
	Activity in Technical Moves	37	5.60	1.7	0	0.00	−1.7			
	Opposition Not Counted	67	10.10	2.3 ^†^	0	0.00	−2.3 ^†^			
	Opposition Counted	182	27.50	4.3 ^†^	0	0.00	−4.3 ^†^			
	Matches of all Kinds	136	20.50	−12.2 ^†^	49	100.00	12.2 ^†^			
GS	Static Activity	23	3.50	1.3	0	0.00	−1.3	22.06 (0.00 *)	24.43 (0.00 *)	0.18
	Small Spaces	68	10.30	2.4 ^†^	0	0.00	−2.4 ^†^			
	Medium Spaces	85	12.80	2.7 ^†^	0	0.00	−2.7 ^†^			
	Large Spaces	34	5.10	1.6	0	0.00	−1.6			
	Repetition of Spaces	452	68.30	−4.7 ^†^	49	100.00	4.7 ^†^			
CI	Individual Intervention	255	38.50	5.4 ^†^	0	0.00	−5.4 ^†^	67.30 (0.00 *)	72.90 (0.00 *)	0.31
	Intervention of 2 Players	65	9.80	2.3 ^†^	0	0.00	−2.3 ^†^			
	Intervention of 3 Players	57	8.60	2.1 ^†^	0	0.00	−2.1 ^†^			
	Intervention of 4 Players	22	3.30	1.3	0	0.00	−1.3			
	Intervention of 5 or +Players	263	39.70	−8.2 ^†^	49	100.00	8.2 ^†^			

Note: *n* = Frequency; *ASR* = Adjusted Standardized Residuals; *X*^2^ = Chi-Square Test; *f* = Fisher’s Exact Test; *Vc* = Cramer’s V Coefficient; DO = Degree of Opposition; DT = Density of the Task; PSP = Percentage of Simultaneous Performers; CL = Competitive Load; GS = Game Space; CI = Cognitive Implication. ^†^
*ASR* > |1.96|; * *p* < 0.05.

**Table 5 sensors-22-02870-t005:** Pairwise comparisons according to the intensity variables.

Intensity	Specific Position		Game Situation	Training Task Type
	Training Session	Official Match	Training Session Only	
eTL*min	-	-	I-SSG*NO/E-SSG*NO/FG*NO/E-SSG*I-SSG/FG*I-SSG	SG*SE/CG*SE/P-S*SE/SP*ES/CG*CE/P-S*CE/SP*CE/CG*SG/P-S*SG/SP*SG/SP*CG
m/min	-	-	FG*I-SSG/FG*NO/FG*E-SSG	SP*SG/P-S*SG/CE*SG/SP*SE/P-S*SE/CE*SE/SP*CG/P-S*CG/CE*CG
Acc/min	S*D/W*D	S*D/W*D	I-SSG*FG/E-SSG*FG/NO*FG/NO*I-SSG /NO*E-SSG	SG*CE/SE*CE/SG*P-S/SE*P-S/SG*SP/SE*SP/SG*CG/SE*CG
Dec/min	S*D/W*D	S*D/W*D	I-SSG*FG/E-SSG*FG/NO*FG/NO*I-SSG /NO*E-SSG	SG*CE/SE*CE/SG*P-S/SE*P-S/SG*SP/SE*SP/SG*CG/SE*CG
HIA/min	-	-	I-SSG*NO/E-SSG*NO/FG*NO/FG*I-SSG	CG*SE/SP*SE/SG*SE/P-S*SE/SP*CE/SG*CG/P-S*CG
Sprints/min	-	-	FG*NO	P-S*CE/P-S*SE/P-S*CG/P-S*SG/P-S*SP
PL/min	W*D/W*S	-	FG*E-SSG/FG*NO/FG*I-SSG	SE*SG/CG*SG/SP*SG/P-S*SG/CE*SG/P-S*SE/P-S*CG
HR_avg_	D*M/D*S/D*W	D*WR	E-SSG*NO/I-SSG*NO/FG*NO/FG*E-SSG/FG*I-SSG	CG*SE/SP*SE/P-S*SE/CG*SG/SP*SG/P-S*SG
Relative HR%	D*M/D*S/D*W	-	I-SSG*NO/FG*NO/FG*E-SSG/FG*I-SSG	CG*SG/SP*SG/P-S*SG/CG*SE/SP*SE/P-S*SE

Note: eTL = external Training Intensity; min = Minute; m = Meter; Acc = Acceleration; Dec = Deceleration; HIA = High-Intensity Activity; PL = Player Load; iTL = internal Training Intensity; HR = Heart Rate; avg = Average; D = Defender; W = Wingback; M = Midfielder; WR = Winger; S = Striker; WO = Without Opposition Task; I-SSG = Inequality Small-Sided Game; E-SSG = Equality Small-Sided Game; FG = Full Game; SE = Simple Exercise; CE = Complex Exercise; SG = Simple Game; CG = Complex Game; P-S = Adapted Game; SP = Sport.

**Table 6 sensors-22-02870-t006:** Inferential results for the intensities according to the type of session.

Intensity		Training Session	Official Match	Inter-Group Differences		
		M ± SD		*U*	*p*	*r*
Subjective eTL ^1^	eTL*min	419.77 ± 218.41	1160.44 ± 298.79	1693.00	0.00 *	0.39
Objective eTL	m/min	57.26 ± 20.95	91.28 ± 11.64	2334.50	0.00 *	0.37
	Acc/min	37.11 ± 7.09	31.15 ± 2.60	8112.50	0.00 *	0.22
	Dec/min	37.09 ± 7.09	31.14 ± 2.61	8134.50	0.00 *	0.22
	HIA/min	0.48 ± 1.64	2.20 ± 1.84	3477.00	0.00 *	0.41
	Sprints/min	1.71 ± 5.96	7.11 ± 14.87	11057.00	0.00 *	0.23
	PL/min	2.33 ± 13.94	1.28 ± 0.24	5392.50	0.00 *	0.29
Objective iTL	HR_avg_	126.47 ± 24.95	152.51 ± 18.30	3037.50	0.00 *	0.28
	Relative HR%	65.97 ± 14.36	80.58 ± 10.03	2869.00	0.00 *	0.29

Note: M = Mean; SD = Standard Deviation; *U* = Mann–Whitney U Test; *r* = Rosenthal’s r Test; eTL = external Training Intensity; min = Minute; m = Meter; Acc = Acceleration; Dec = Deceleration; HIA = High-Intensity Activity; PL = Player Load; iTL = internal Training Intensity; HR = Heart Rate; avg = Average. ^1^ The subjective eTL calculated using SIATE observation sheet. * *p* < 0.05.

## Data Availability

Data will be available upon reasonable request to the corresponding author.

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
