# Peer review of "Analysis of Intensities Using Inertial Motion Devices in Female Soccer: Do You Train like You Compete?"

_sensors, 2022, doi:10.3390/s22082870_

Round 1
Reviewer 1 Report
Comparing variables of training sessions with game situation by SIATE, seems useless for the study.
SIATE can be useful for characterizing training sessions, but completely ineffective for studying game situations.
But it's good work
Author Response
Dear reviewer,
We would like to express our gratitude to you for the time in reviewing our manuscript.
Kind regards.
Reviewer 2 Report
ABSTRACT
Lines 16-17: “moreover” does not make sense based on the previous sentence.
Line 17: add the study design type and the total sample (matches and training sessions analyzed).
Line 17: add the main outcomes and the instruments used to obtain them
Lines 28-42: maybe too long, and without a solid statistical presentation. Would be better to improve the study objectives and answer the research question, point-by-point in the results.
Lines 42-43: this is not a conclusion from the study. It seems to be a wish or a projection, but not studied in the current research. Maybe better to remove and replace for a practical implication coming from the results.
INTRODUCTION
Line 47: increasing by how much?
Lines 51-52: the link between ideas is not evident or makes sense. This should be clearly improved.
Line 58: would be interesting to add some examples about task constraints used by coaches to manage the training impact on players.
Line 58-59: again, no link between paragraphs. The rationale must be improved and the paragraphs must follow in a logical and integrated fashion.
Line 60: avoid using the term “workload”. Please consider the recommendations included in this article: Staunton, C. A., Abt, G., Weaving, D., & Wundersitz, D. W. (2021). Misuse of the term ‘load’ in sport and exercise science. Journal of Science and Medicine in Sport.
Lines 59-71: introduction to the measures has been based on a non-objective approach. My recommendation is that can detail the instruments and different measures possible to obtain and after that, choosing those will be analyzed later in the article.
Line 79: summarize the main evidence about the locomotor and physiological impact of different task constraints while playing small-sided games.
Line 91: present the main methodological characteristics of the SIATE and the related work about the use of this system.
Line 97: the study objective appears without a statement of contribution or a rationale. Work on the statement of contribution to justify the objective of the study.
METHODS
Line 108: Describe the setting, locations, and relevant dates, including periods of recruitment, exposure, follow-up, and data collection. Maybe a timeline could be added in form of a table or figure with specific days of data collection and contextualization based on the period of the season. Moreover, add a range of environmental conditions during data collection and the number of players analyzed for each training session and match.
Line 113: Give the eligibility criteria and the sources and methods of selection of participants. Consider reporting how data was collected considering repeated measures and the range of data collected on average and per player.
Table 1: replace weight by body mass and height by stature.
Line 122: contextualize the table. The data in training was organized for the type of exercises? Who made the organization? What about the reliability of the observer to classify? And the expertise?
Line 150: add the technical characteristics of the device, and the reliability of the instrument to measure each of the main outcomes. Moreover, in the end, add the main outcomes extracted and the information if was standardized per time or not.
Lines 165-168: why use both parametric and non-parametric? Which indicators were used to used parametric for some cases and non-parametric for others? Add a table with normality and homogeneity values per outcome. Moreover, add the statistical approaches used for the case of missing cases (repeated measures).
RESULTS
Figure 1: add the pairwise comparisons (significant differences)
DISCUSSION
Line 222: synthesize the evidence found. Must be objective and provide a solid picture of the main findings.
Line 333: add future research ideas.
Author Response
Dear reviewer,
First of all, we would like to express our gratitude to you for the time in reviewing our manuscript and for providing us comments helpful to improve this manuscript quality. We have found suggestions very constructive and have answered their concerns.
--------------------
All manuscript
- All corrections were marked in red.
--------------------
Reviewer’ note: Lines 16-17: “moreover” does not make sense based on the previous sentence.
Authors’ response: Thank you for your suggestion. The "moreover" connector has been removed.
--------------------
Reviewer’ note: Line 17: add the study design type and the total sample (matches and training sessions analyzed).
Authors’ response: Based on your comment, the study design and total sample (training sessions and matches analyzed) have been specified (lines 17 to 19).
--------------------
Reviewer’ note: Line 17: add the main outcomes and the instruments used to obtain them.
Authors’ response: The instruments used for data collection are shown in the abstract.
--------------------
Reviewer’ note: Lines 28-42: maybe too long, and without a solid statistical presentation. Would be better to improve the study objectives and answer the research question, point-by-point in the results.
Authors’ response: We agree with your comment. Thus, we have linked each part of the study with the statistical tests measuring each variable and the results obtained.
--------------------
Reviewer’ note: Lines 42-43: this is not a conclusion from the study. It seems to be a wish or a projection, but not studied in the current research. Maybe better to remove and replace for a practical implication coming from the results.
Authors’ response: Based on your comment, the sentence was replaced by a practical implication (lines 43 to 44).
--------------------
Reviewer’ note: Line 47: increasing by how much?
Authors’ response: The justification for the increase in female's soccer is in the following sentence. In addition, data have been updated (lines 48 to 50).
--------------------
Reviewer’ note: Lines 51-52: the link between ideas is not evident or makes sense. This should be clearly improved.
Authors’ response: Thank you for your suggestion. The wording of the paragraph has been modified (line 53).
--------------------
Reviewer’ note: Line 58-59: again, no link between paragraphs. The rationale must be improved and the paragraphs must follow in a logical and integrated fashion.
Authors’ response: Based on your comment, the wording of the paragraph has been modified (lines 62-63).
--------------------
Reviewer’ note: Line 60: avoid using the term “workload”. Please consider the recommendations included in this article: Staunton, C. A., Abt, G., Weaving, D., & Wundersitz, D. W. (2021). Misuse of the term ‘load’ in sport and exercise science. Journal of Science and Medicine in Sport.
Authors’ response: The term "workload" has been replaced by "intensity" in all manuscript.
--------------------
Reviewer’ note: Lines 59-71: introduction to the measures has been based on a non-objective approach. My recommendation is that can detail the instruments and different measures possible to obtain and after that, choosing those will be analyzed later in the article.
Authors’ response: First, the objective measures for quantifying intensities have been described (lines 63 to 76). The, the subjective measures have been described (lines 92 to 103).
--------------------
Reviewer’ note: Line 79: summarize the main evidence about the locomotor and physiological impact of different task constraints while playing small-sided games.
Authors’ response: Thank you for your suggestion. Information on the physical-physiological impact of SSG constraints has been added (lines 84 to 87).
--------------------
Reviewer’ note: Line 91: present the main methodological characteristics of the SIATE and the related work about the use of this system.
Authors’ response: Based on your comment, the main methodological characteristics of SIATE and several works related to the use of this system have been indicated. (lines 95 to 97).
--------------------
Reviewer’ note: Line 97: the study objective appears without a statement of contribution or a rationale. Work on the statement of contribution to justify the objective of the study.
Authors’ response: We agree with your comment. Thus, the objective of the study has been justified (lines 109 to 111).
--------------------
Reviewer’ note: Line 108: Describe the setting, locations, and relevant dates, including periods of recruitment, exposure, follow-up, and data collection. Maybe a timeline could be added in form of a table or figure with specific days of data collection and contextualization based on the period of the season. Moreover, add a range of environmental conditions during data collection and the number of players analyzed for each training session and match.
Authors’ response: Thank you for your suggestion. Figure 1 has been added, showing the specific days of data collection and contextualization (line 123).
--------------------
Reviewer’ note: Line 113: Give the eligibility criteria and the sources and methods of selection of participants.
Authors’ response: Based on your suggestion, the criteria for selection of study participants were indicated (lines 128 to 129).
--------------------
Reviewer’ note: Table 1: replace weight by body mass and height by stature.
Authors’ response: Thank you for your comment, both terms have been replaced.
--------------------
Reviewer’ note: Line 122: contextualize the table. The data in training was organized for the type of exercises? Who made the organization? What about the reliability of the observer to classify? And the expertise?
Authors’ response: Based on your comment, Table 2 has been contextualized.
--------------------
Reviewer’ note: Line 150: add the technical characteristics of the device, and the reliability of the instrument to measure each of the main outcomes. Moreover, in the end, add the main outcomes extracted and the information if was standardized per time or not.
Authors’ response: Information on the characteristics of Wimu ProTM device and articles demonstrating their validity and reliability for sports analysis have been added. (lines 168 to 172).
--------------------
Reviewer’ note: Lines 165-168: why use both parametric and non-parametric? Which indicators were used to used parametric for some cases and non-parametric for others? Add a table with normality and homogeneity values per outcome. Moreover, add the statistical approaches used for the case of missing cases (repeated measures).
Authors’ response: Thank you for your comment, but we consider that the article has an excessive number of tables. Therefore, in all figures and tables we have been indicated the tests used (not section) to help the reader.
--------------------
Reviewer’ note: Figure 1: add the pairwise comparisons (significant differences).
Authors’ response: Based on your suggestion, Table 5 has been added with pairwise comparisons (significant differences) (line 242).
--------------------
Reviewer’ note: Line 222: synthesize the evidence found. Must be objective and provide a solid picture of the main findings.
Authors’ response: Thank you for your suggestion. The evidence found have been synthesized (lines 265 to 268).
--------------------
Reviewer’ note: Line 333: add future research ideas.
Authors’ response: Thank you for your comment. Future research ideas have been added (lines 375 to 378).
Kind regards.
Reviewer 3 Report
This is a very original and meticulous work on the study and comparative investigation of workload in female soccer is limited. The authors classify the training processes with quantitative findings, quite appropriately and use the proper statistical tools to evaluate them.
I find the paper quite original and strong in that one can deduct conclusions and enhance training strategy according to these findings. Also, the study concerns female soccer for which data are less.
It would be interesting to further read, at the conclusions, of how such methods can be improved by finding further causal relations that are not predicted and may be discovered.
Author Response
Dear reviewer,
First of all, we would like to express our gratitude to you for the time in reviewing our manuscript. We have found your suggestion very constructive and have answered you concern.
--------------------
All manuscript
- All corrections were marked in red.
--------------------
Reviewer’ note: It would be interesting to further read, at the conclusions, of how such methods can be improved by finding further causal relations that are not predicted and may be discovered.
Authors’ response: Thank you for your comment. The conclusions have been completed with new information.
Kind regards.
Round 2
Reviewer 2 Report
The article was improved and can be accepted in the current form.